# Development of Miconazole-Loaded Microemulsions for Enhanced Topical Delivery and Non-Destructive Analysis by Near-Infrared Spectroscopy

**DOI:** 10.3390/pharmaceutics15061637

**Published:** 2023-06-01

**Authors:** Chutima Phechkrajang, Wichuda Phiphitphibunsuk, Rapee Sukthongchaikool, Nantana Nuchtavorn, Jiraporn Leanpolchareanchai

**Affiliations:** 1Department of Pharmaceutical Chemistry, Faculty of Pharmacy, Mahidol University, Bangkok 10400, Thailand; chutima.mat@mahidol.ac.th (C.P.); rapee.suh@mahidol.ac.th (R.S.); nantana.nuc@mahidol.edu (N.N.); 2School of Pharmaceutical Sciences, University of Phayao, Phayao 56000, Thailand; wichuda.ph@up.ac.th; 3Department of Pharmacy, Faculty of Pharmacy, Mahidol University, Bangkok 10400, Thailand

**Keywords:** anticandidal activity, miconazole nitrate, microemulsion, near-infrared spectroscopy, skin permeation, topical delivery

## Abstract

The antifungal drug miconazole nitrate has a low solubility in water, leading to reduced therapeutic efficacy. To address this limitation, miconazole-loaded microemulsions were developed and assessed for topical skin delivery, prepared through spontaneous emulsification with oleic acid and water. The surfactant phase included a mixture of polyoxyethylene sorbitan monooleate (PSM) and various cosurfactants (ethanol, 2-(2-ethoxyethoxy) ethanol, or 2-propanol). The optimal miconazole-loaded microemulsion containing PSM and ethanol at a ratio of 1:1 showed a mean cumulative drug permeation of 87.6 ± 5.8 μg/cm^2^ across pig skin. The formulation exhibited higher cumulative permeation, permeation flux, and drug deposition than conventional cream and significantly increased the in vitro inhibition of *Candida albicans* compared with cream (*p* < 0.05). Over the course of a 3-month study conducted at a temperature of 30 ± 2 °C, the microemulsion exhibited favorable physicochemical stability. This outcome signifies its potential suitability as a carrier for effectively administering miconazole through topical administration. Additionally, a non-destructive technique employing near-infrared spectroscopy coupled with a partial least-squares regression (PLSR) model was developed to quantitatively analyze microemulsions containing miconazole nitrate. This approach eliminates the need for sample preparation. The optimal PLSR model was derived by utilizing orthogonal signal correction pretreated data with one latent factor. This model exhibited a remarkable R^2^ value of 0.9919 and a root mean square error of calibration of 0.0488. Consequently, this methodology holds potential for effectively monitoring the quantity of miconazole nitrate in various formulations, including both conventional and innovative ones.

## 1. Introduction

The imidazole family member miconazole nitrate has broad-spectrum antifungal activity by damaging the fungal cell membranes, which leads to the lysis of their cell walls [1]. Due to its limited solubility (<1 μg/mL) and substantial hepatic alteration, it has little systemic effectiveness [2]. Miconazole nitrate has been shown to be a potent topical antifungal agent against dermatophytes and yeasts. Therefore, miconazole-based preparations are presently utilized for topical administration and are commercially accessible in a range of topical formulations, including creams, lotions, liquid sprays, and suppositories specifically designed for vaginal application [3]. According to a previous investigation [4], the effectiveness of miconazole in treating cutaneous diseases via topical applications was found to be limited by inadequate skin penetration. Consequently, there is a need for an innovative approach that can localize the drug to the targeted area to enhance the therapeutic efficacy. This approach should deviate from traditional dosage forms and employ a topical mode of delivery. To address these challenges, researchers have explored the use of nanotechnology-based nanocarriers, such as liposomes, ethosomes, and niosomes [5,6,7]. However, these vesicular carrier systems have not been able to achieve the desired outcome of completely eradicating infections from the viable epidermis due to issues of physical, chemical, and storage instability within the formulations. Therefore, a microemulsion system was selected for this study due to its ease of preparation, ability to incorporate hydrophobic drug, stability, and most importantly, its greater physical stability in plasma compared to other vesicular systems. A microemulsion refers to a stable and homogeneous system comprising water, oil, and surfactant. It exhibits isotropic properties and is typically transparent or translucent. This formulation commonly incorporates a cosurfactant and has gained attention as a promising drug delivery system. Its advantageous characteristics include enhancing the solubility and bioavailability of drugs that possess limited water solubility [8,9].

In previous studies, microemulsion has been employed as a carrier for the topical delivery of miconazole. Ofokansi et al. [10] compared the in vitro antifungal effectiveness of a miconazole nitrate-loaded microemulsion stabilized with poloxamer and a commercial topical miconazole (Fungusol^®^) solution against clinically isolated *Candida albicans*. The new formulation showed an increase in in vitro antifungal activity, indicating that poloxamer-stabilized microemulsion may be a potential vehicle for improving topically administered miconazole nitrate. Later, Shahzadi et al. [11] evaluated the in vitro antifungal activity against *C. albicans* and drug release profiles of a miconazole nitrate-loaded microemulsion and a reference cream. The microemulsion preparation showed superior results in in vitro antifungal activity, drug release, and flux compared to the cream preparation. Sadique et al. [12] also compared the in vitro antifungal effectiveness against *C. albicans* of a miconazole nitrate-loaded microemulsion and 1 and 2% commercially available miconazole nitrate formulations (Miconit^®^). The final selected microemulsion formulation showed enhanced antifungal activity compared to the commercially available formulations, and it was concluded that microemulsion is a promising carrier system for topical miconazole nitrate delivery.

The aim of this research was to create and evaluate a microemulsion formulation as a topical delivery system to enhance the delivery of miconazole nitrate. The microemulsions containing miconazole nitrate were prepared by employing the spontaneous emulsification method, with oleic acid and deionized (DI) water used as oil and aqueous phases, respectively. This study investigated the effects of the cosurfactant type and weight ratio of surfactant and cosurfactant mixtures (K_m_) on the preparation and physicochemical characteristics of the obtained microemulsion formulations, which differed from the previous studies [10,11,12]. The surfactant phase was made up of a mixture of polyoxyethylene sorbitan monooleate (PSM) and various cosurfactants (ethanol, 2-(2-ethoxyethoxy) ethanol (DEGEE), or 2-propanol) at weight ratios of 1:1, 2:1, and 3:1. The formulations’ physicochemical properties, such as physical appearance, conductivity, pH, viscosity and rheological behavior, and particle size, were examined. The selected microemulsion’s influence on ex vivo skin permeation and in vitro antifungal activity against *C. albicans* was assessed and compared with a conventional cream. The formulation’s physicochemical stability was examined by storing it at room temperature (30 ± 2 °C) for a period of 3 months.

Furthermore, a non-destructive analysis utilizing near-infrared (NIR) spectroscopy was created for quantifying miconazole nitrate-loaded microemulsion formulations. In the pharmaceutical industry, NIR spectroscopy is a widely used technique because it provides valuable insights into the chemical composition and physical properties of samples [13]. NIR is widely used in conjunction with fiber optical probes to identify raw materials. Furthermore, several studies have suggested the use of NIR spectroscopy to analyze drug substances and excipients quantitatively within pharmaceutical formulations through chemometric multivariate calibration and partial least-squares regression (PLSR) [14,15]. Although the literature extensively covers quantitative NIR analyses of solid dosage forms, there has been relatively less focus on the analysis of liquid formulations [14]. In this study, six PLSR models for the quantitative determination of miconazole nitrate-loaded microemulsions were developed using benchtop NIR spectroscopy with reference values from high-performance liquid chromatography (HPLC) method. To our understanding, this represents the initial establishment of a benchtop NIR spectroscopic method for quantifying miconazole nitrate-loaded microemulsion.

## 2. Materials and Methods

### 2.1. Materials and Chemicals

Miconazole nitrate USP (≥99%) was sourced from Gufic Biosciences Limited (Mumbai, India). Oleic acid USP XVI was acquired from Union Chemical 1986 Co., Ltd. (Bangkok, Thailand). Caprylic/capric triglyceride was obtained from Sasol Olefins & Surfactants GmbH (Hamburg, Germany). PSM and sorbitan monooleate (SM) were purchased from S. Tong Chemicals Co., Ltd. (Bangkok, Thailand). DEGEE was sourced from Sigma Chemical Company (St. Louis, MO, USA). Additionally, 2-propanol and ethanol (95% *v*/*v*) were purchased from RCI Labscan Limited (Samutsakorn, Thailand). Coomassie brilliant blue R-250 was procured from Bio-Rad Laboratories (Hercules, CA, USA), while phosphate buffered saline (PBS) tablets utilized for the preparation of pH 7.4 buffer solution were acquired from Sigma Chemical Company (St. Louis, MO, USA). Sabouraud dextrose agar (SDA) was obtained from Scharlab, S.L. (Barcelona, Spain). Methanol (HPLC grade) was sourced from Honeywell International Inc. (Seoul, Republic of Korea). During the course of the experiments, DI water was utilized. Additionally, other chemicals and reagents of analytical grade or higher were procured from regional suppliers and utilized without the need for additional purification.

### 2.2. Selection of Microemulsion Components

The miconazole nitrate solubility in different oils, such as oleic acid and caprylic/capric triglyceride; surfactants, including PSM and SM; and cosurfactants, i.e., ethanol, DEGEE, and 2-propanol, was assessed using the shake-flask method [16]. Separately, 2 g of oil, surfactant, or cosurfactant was placed into stoppered vials with a capacity of 5 mL, and an excess amount of miconazole nitrate was added to each vial. To achieve equilibrium, the sample mixtures underwent mechanical stirring in a shaking water bath set at a rate of 100 strokes per minute (spm) for a duration of 72 h at a temperature of 30 ± 2 °C. The equilibrated samples were centrifuged at a speed of 9503× *g* for a duration of 10 min. The resulting clear supernatants were carefully collected and diluted with suitable solvents. Once filtered through a polytetrafluoroethylene (PTFE) membrane filter with a diameter of 0.45 µm, the filtrates were subjected to analysis using HPLC. This solubility testing procedure was repeated three times. The oil and surfactant that exhibited the maximum solubility of miconazole nitrate were employed for further studies.

### 2.3. Construction of Pseudo-Ternary Phase Diagram

To construct pseudo-ternary phase diagrams for unloaded microemulsions, oleic acid and PSM were chosen as the oil and surfactant, respectively, based on solubility data. The surfactant phase consisted of PSM combined with different cosurfactants (ethanol, DEGEE, or 2-propanol) at weight ratios of 1:1, 2:1, and 3:1, while the aqueous phase comprised DI water. For each phase diagram, the surfactant mixture was put into the oil phase, resulting in weight ratios of 9:1, 8:2, 7:3, 6:4, 5:5, 4:6, 3:7, 2:8, and 1:9 for the surfactant mixture to oleic acid. A magnetic stirrer was used to thoroughly mix these mixtures until a homogenous dispersion was achieved. Following the addition of the aqueous phase, various concentrations ranging from 0 to 90% in 10% weight intervals were obtained. The systems were agitated for 5 min using a magnetic stirrer before being kept at 30 ± 2 °C for a duration of 24 h to establish equilibrium. The resulting formulations exhibited transparent properties and were classified as microemulsions. Phase separation was evaluated by visual observation of turbidity. In order to determine the presence of microemulsions, gels, or two-phase regions, additional visual observations were performed. The type of microemulsion was then identified through a dilution test using either oleic acid or a brilliant blue aqueous solution. It was expected that the water-in-oil (w/o) microemulsions would exhibit miscibility with oleic acid, but immiscibility with the brilliant blue aqueous solution. Conversely, the oil-in-water (o/w) microemulsions were predicted to display opposite characteristics. The microemulsion area of the system was built on a triangle graph using SigmaPlot^®^ 10.0 software. The extent to which the microemulsion covered the overall percentage of the phase diagram was determined using a cut-and-weigh method [17]. However, no efforts were made to ascertain the regions associated with other structural arrangements.

### 2.4. Preparation of Miconazole Nitrate-Loaded Microemulsions

Regarding the phase diagram’s microemulsion area, blank microemulsion formulations were selected and assessed for their ability to solubilize miconazole nitrate in order to evaluate the highest drug loading capacity. Two grams of blank microemulsions or water were placed in separate 5 mL capacity stoppered vials together with an excess amount of miconazole nitrate. Until equilibrium was reached, the sample mixtures were subjected to mechanical stirring in a shaking water bath running at a speed of 100 spm for 72 h at a temperature of 30 ± 2 °C. Following equilibration, the samples were centrifuged at 9503× *g* for 10 min, resulting in clear supernatants. These supernatants were diluted with proper solvents and then filtered through a PTFE membrane filter. The filtrates were subsequently analyzed using HPLC. The solubility tests were conducted three times.

Following the determination of miconazole nitrate’s solubility in blank microemulsions, the drug was accurately weighed and loaded into each unloaded microemulsion base using a magnetic stirrer until a homogenous mixture was achieved. Following that, the created miconazole nitrate-loaded microemulsions at a 1% *w*/*w* concentration were maintained at a temperature of 30 ± 2 °C for a period of 24 h to reach equilibrium before more testing. Subsequently, the microemulsions were subjected to characterization and compared with their corresponding blank formulations.

### 2.5. Characterization of Miconazole Nitrate-Loaded Microemulsions

#### 2.5.1. Macroscopic Observation

The physical appearance of all samples was visually evaluated, taking into consideration factors such as clarity, color, and homogeneity. Samples were inspected with a cross-polarized light microscope (Eclipse E200 Microscope, Nikon Corporation Instruments Company, Tokyo, Japan) at a magnification of 10 × 100 to confirm the microemulsion’s isotropic character. A drop of the sample was put between a cover slip and a glass slide and examined under cross-polarized light. These experiments were conducted at 30 ± 2 °C.

#### 2.5.2. Conductivity and pH Measurement

To determine whether the samples were oil-continuous or aqueous-continuous microemulsions, the electrical conductivity of the samples was assessed using a conductivity tester (ECTestr^TM^ 11, Eutech Instruments Pte. Ltd., Singapore), and the results were compared against the conductivity values of both oleic acid and water. The apparent pH values of the samples were determined using a pH meter (CyberScan pH110, Eutech Instruments Pte. Ltd.). At 30 ± 2 °C, each measurement was made in triplicate.

#### 2.5.3. Viscosity and Rheological Behavior Measurement

Using a rotational rheometer equipped with a cylindrical stainless steel measurement system (ϕ 34 mm ISO 3219 Z34 DIN, HAAKE^TM^ RotoVisco^TM^ 1, Thermo Fisher Scientific GmbH, Dreieich, Germany), the viscosity and rheological behavior of the samples were assessed at various shear rates. The rheological data obtained from the viscometer were driven and calculated using the HAAKE RheoWin 4 software. To maintain a temperature of 30 ± 2 °C, a Thermo Scientific HAAKE EZ Cool 80 circulator was utilized. The sample volume for the tests was 40 mL. By building a rheogram plotting shear stress against shear rate, the rheological behavior of the systems was investigated. The experiments yielded the apparent viscosity data at a shear rate of 1000 s^−1^ and a controlled temperature of 30 ± 2 °C. The tests were carried out in triplicate.

#### 2.5.4. Particle Size Measurement

The formulations were subjected to analysis using a Zetasizer Nano-ZS instrument (Malvern Instruments Ltd., Worcestershire, UK) to determine the average droplet size (z-ave) and polydispersity index (PDI). The software automatically calculated the measurement position inside the cuvette, and the measurements were taken at a detection angle of 173°. The Zetasizer software used the Stokes–Einstein relationship to determine the sample droplet size. The studies were done in triplicate at 30 ± 2 °C without dilution of the samples.

### 2.6. Thermodynamic Stability of Miconazole Nitrate-Loaded Microemulsions

The samples underwent a centrifugation test using a centrifuge (Hettich Universal 30F, Andreas Hettich GmbH & Co. KG, Osterode am Harz, Germany) at 21,382× *g* for 1 h at a temperature of 30 ± 2 °C. The degree of phase separation, visually evaluated, was used as an indication of the physical stability of the formulations following centrifugation. Only the formulations that demonstrated no phase separation were considered for the freeze–thaw stress test.

The formulations underwent three complete cycles of freezing at −20 °C for 24 h followed by thawing at 30 ± 2 °C for 24 h each. The assessment of physical instability, based on the degree of phase separation, was conducted after the completion of the third cycle.

### 2.7. Ex Vivo Skin Permeation Study of the Selected Miconazole Nitrate-Loaded Microemulsion

Through a newborn pig skin membrane, miconazole nitrate from the chosen microemulsion and conventional cream (1% *w*/*w* miconazole nitrate cream) was examined ex vivo. Thermo-regulated vertical Franz diffusion cells were employed, maintaining a temperature of 37 ± 1 °C to achieve a skin surface temperature of 32 ± 1 °C. Since miconazole nitrate has limited water solubility (<1 μg/mL), the use of any additive to enhance the drug release under the sink condition is necessary. In this study, a mixture of PBS (0.01 M, pH 7.4) and methanol at a ratio of 80 and 20 (%*v*/*v*) was used as a receptor medium in order to enhance the drug solubility and achieve the sink condition during the experiment. In addition, the medium was proven to maintain the integrity of the pig skin, and the use of hydroalcoholic solutions (e.g., methanol, ethanol) as the receptor medium has been accepted and applied in several other publications [18,19,20,21]. Ten milliliters of the medium were degassed before use. The skin membrane used in the study was taken from the abdominal area of newborn pigs, which were sourced from a regional pig farm located in Nakhon Pathom Province, Thailand, which operates under the supervision of the Department of Livestock Development. The newborn pigs weighed between 1.4 and 1.8 kg and died naturally shortly after birth. As this source of skin is categorized as slaughter waste, it is exempt from the ethics committee of the Institute for Animal Care and Use Committee, Faculty of Pharmacy, Mahidol University. In addition, the skins were collected from carcasses, whereas the ethics committee concerns only alive animals, as indicated by the definition of “animals” in “The Animals for Scientific Purposes Act, B.E. 2558 (A.D. 2015), Thailand” [22]. Consent was verbally obtained from slaughterhouse owners to collect all of the samples from carcasses. The epidermal hair in the abdominal area was clipped with an electric hair clipper. The hair was carefully removed as close as possible to the skin without damaging it. The skin was then rinsed under running tap water in order to remove hair from the surface, and then excised with a scalpel and a number 24 surgical blade. The subcutaneous fat and underlying tissues were carefully removed from the dermal surface. Subsequently, the skin pieces were rinsed with PBS, blotted dry with paper towels, and carefully wrapped in aluminum foil. The prepared membranes were then stored at −20 °C until further use [23]. Before mounting the membrane on the receptor compartment, it was allowed to thaw and hydrate by soaking it in PBS overnight at room temperature (30 ± 2 °C). The membrane was arranged in a manner where the stratum corneum was oriented in an upward-facing position. For the experiments, a sample weighing 0.5 g was carefully applied onto the skin membrane, covering an effective area of 1.77 cm^2^. At specified time intervals (0.5, 1, 2, 4, 6, 12 and 24 h), a volume of 0.5 mL of receptor fluid was withdrawn, and an equal volume of fresh receptor fluid was immediately added. The collected samples underwent filtration using a PTFE membrane filter. Subsequently, the filtrates were subjected to HPLC analysis to determine the amount of active permeation. The cumulative permeated amount (Q_t_) of miconazole nitrate was determined using Equation (1).
(1)$$Q_{t}=V_{r}C_{t}+\sum_{i=0}^{t–1}V_{s}C_{i}$$
where C_t_ is the active compound concentration in the receptor fluid at each sampling time, C_i_ is the active compound concentration of the i-th sample, and V_r_ and V_s_ are the volumes of the receptor fluid and the sample, respectively. The permeation profiles were generated by plotting the cumulative quantity of miconazole nitrate that permeated through the skin membrane per unit area of the membrane over time. The permeation rates were then determined by interpolating the permeation profiles using linear regression.

After the skin permeation study, the miconazole nitrate that had remained in the skin membrane was extracted using the previously established method [24]. The extraction procedure involved wiping the surface of each skin membrane with cotton soaked in a specific volume of PBS to remove any residual sample. Subsequently, the skin was cut into small pieces, homogenized in methanol, and filtered. The skin permeation and retention studies were done at least three times using skin samples from a minimum of three newborn pigs.

### 2.8. In Vitro Antifungal Activity of the Selected Miconazole Nitrate-Loaded Microemulsion

Antifungal activities of the selected miconazole nitrate-loaded microemulsion and conventional cream (1% *w*/*w* miconazole nitrate cream) were evaluated against the standard strain *C. albicans* ATCC 10231 using the cup-plate method [25]. The SDA medium was prepared by weighing the required SDA amount and dissolving it in water, followed by sterilization in an autoclave at 121 °C for 15 min. Once the melted nutrient medium had cooled to 60 °C under sterile conditions, it was poured into Petri dishes. The dishes were then inoculated with the microorganism being tested. After the plate had a chance to solidify, a sterile borer was used to create a well. Each well received the tested samples, which were then left to stand for 1 h. After that, the plates were placed in an incubator (Binder, Inc., Bohemia, NY, USA) and kept at 37 ± 1 °C. Using a graduated scale, the zone of inhibition (ZOI, mm) was determined after 24 h. The results were reported as mean ± standard deviation (SD), with the test being run in triplicate.

### 2.9. Stability Study of the Selected Miconazole Nitrate-Loaded Microemulsion

After storing the selected miconazole nitrate-loaded microemulsion at room temperature (30 ± 2 °C) for a duration of 3 months, its stability was assessed. The microemulsion formulation was collected at predetermined intervals, including immediately following preparation and at 1, 2, and 3 months, to evaluate its physical appearance, including color, phase separation, and clarity. Utilizing HPLC, the formulation’s miconazole nitrate chemical stability was evaluated. At 30 ± 2 °C, each measurement was carried out three times.

### 2.10. Analysis of Miconazole Nitrate by HPLC

A Shimadzu HPLC system (Shimadzu Scientific Instruments, Kyoto, Japan) outfitted with a DGU-20A5 degasser, LC-20AD pumping system, SIL-20AHT autosampler, and SPD-20A UV/VIS detector was used to accomplish the quantitative analysis of miconazole nitrate. As the stationary phase, a guard column-equipped Hypersil GOLD^TM^ column (C18; 150 × 4.6 mm, particle size 5 μm; Thermo Fisher Scientific Inc., Carlsbad, CA, USA) was used. Elution was performed at room temperature (30 ± 2 °C) using a mobile phase solution composed of 0.5% *w*/*v* ammonium acetate and methanol (20:80, *v*/*v*) at a flow rate of 1 mL/min. UV detection was done at 263 nm with a sample injection volume of 20 μL. In accordance with the International Conference on Harmonization (ICH) guidelines: Text and methodology (Q2(R1)) [26], the HPLC method was validated to access the analytical processes for linearity, precision, accuracy, limit of detection (LOD), and limit of quantitation (LOQ). Utilizing three series of five different working standard solution concentrations, the linearity of the method was assessed over concentrations ranging from 1 to 20 μg/mL. The standard curve’s slope, y-intercept, and regression coefficient (*r*) were determined. By examining three replications of three distinct spiking standard concentrations (1.5, 7.5, and 15 μg/mL) and computing the percent recovery, accuracy was assessed. Both intra-day (repeatability) and inter-day precision were evaluated to determine precision. Triplicate measurements of three different concentrations (1.5, 7.5, and 15 μg/mL) taken within a day were used to assess the repeatability, whereas these same measurements taken on three different days were used to assess the inter-day precision. A calculation was made to determine the relative standard deviation percentage (%RSD). A signal-to-noise ratio of 3 was used to determine the LOD, while a ratio of 10 was used to calculate the LOQ.

### 2.11. Quantitative Determination of Miconazole Nitrate in Microemulsions by NIR

#### 2.11.1. Preparation of Calibration and Validation Samples

Five concentrations of miconazole nitrate-loaded microemulsion (0.00, 0.50, 0.75, 1.00, 1.25, and 1.50% *w*/*w*) were prepared. Each microemulsion concentration was further divided into 10 sub-samples, resulting in a total of 60 sub-samples. All sub-samples were quantitatively determined for their actual concentration by using the HPLC method described in Section 2.10, and their NIR spectra were obtained as per the condition described in Section 2.11.2.

#### 2.11.2. NIR Spectroscopic Measurement

The NIR spectra of all sub-samples and 10 placebo samples were collected using the NIRFlex^®^ N-500 NIR spectroscope (BÜCHI Labortechnik AG, Flawil, Switzerland) in transflectance mode. The NIRFlex Solids measurement cell, an add-on for measurements of various sample types ranging from solids to liquids, was utilized for microemulsion samples. About 5 mL of microemulsion was placed in a Petri dish and covered with the reference plate, and the NIR spectrum was collected with the resolution of 4 nm between 4000–10,000 cm^−1^ (1000–2500 nm). Duplicate measurements were performed for each sub-sample, and the average NIR spectrum was chosen for further investigation.

#### 2.11.3. PLSR Modelling

The 42 out of 60 sub-samples were randomly selected and combined with 10 placebo samples for use as calibration set samples. The remaining 18 sub-samples were designated as the validation set. The raw NIR spectral data were analyzed using the Unscrambler^®^ program (Aspen Tech, Bedford, MA, USA) to construct the PLSR determination model. Various pretreatment methods, including first (1D) and second (2D) derivatives, standard normal variate (SNV), area normalization, and orthogonal signal correction (OSC), were applied to the raw NIR spectral data. Multiple PLSR models were constructed using raw spectral data and pretreated data, taking wavelength selection into account to obtain the optimum model. Model parameters, such as R^2^ model, root mean square error of calibration (RMSEC, Equation (2)), R^2^ Pearson, root mean square error of prediction (RMSEP, Equation (3)), and bias (Equation (4)), were considered to select the most optimal PLSR model.
(2)$$\mathrm{RMSEC}\;=\sqrt{\frac{\sum_{i=1}^{n}{\left(y_{i}–{\hat{y}}_{i}\right)}^{2}}{n}}$$
(3)$$\mathrm{RMSEP}\;=\sqrt{\frac{\sum_{i=1}^{n}{\left(y_{i}–{\hat{y}}_{i}\right)}^{2}}{n}}$$
(4)$$\mathrm{Bias}\;=\frac{\left(y_{\mathrm{ref}}–y_{\mathrm{pred}}\right)}{y_{\mathrm{ref}}}$$
where y_i_ and ŷ_i_ are the reference and predicted values for sample i from the validation set, respectively, and n is the number of samples in the validation set. The highest R^2^ values and lowest error parameters, such as RMSEC, RMSEP, and bias, were model selection criteria.

### 2.12. Data Analysis

The results were presented as the mean ± SD for three replicates. By utilizing SPSS Statistics 21.0 (IBM, Armonk, NY, USA), the statistical analysis was carried out. One-way analysis of variance (ANOVA) was used to assess significant differences (*p* < 0.05) between the means. Tukey’s honesty significant difference test or Dunnett’s T3 test for multiple comparisons was then used to analyze those differences.

Data analysis for the HPLC experiment and method validation were performed by Excel program. For PLSR modelling of NIR measurement, all calculations were performed with the Unscrambler^®^ program.

## 3. Results and Discussion

### 3.1. Selection of Microemulsion Components

In order to formulate a microemulsion for effective topical delivery of the weakly water-soluble drug miconazole nitrate, it is important to properly choose the oil phase. The selection of components for formulating the microemulsion was based on the miconazole nitrate solubility in the various oils, surfactants, and cosurfactants. Table 1 displays the equilibrium solubility values. Oleic acid exhibited the greatest solubility for miconazole nitrate (0.012 ± 0.003% *w*/*w*) among the various oils, followed by caprylic/capric triglyceride. Among the surfactants, PSM showed the maximum solubility (0.654 ± 0.066% *w*/*w*), followed by SM. For the cosurfactants, 95% ethanol showed the highest solubility (1.164 ± 0.026% *w*/*w*), followed by DEGEE and 2-propanol, respectively. Based on the solubility study of miconazole nitrate, oleic acid and PSM could be the most suitable oil and surfactant for microemulsion development.

### 3.2. Construction of Pseudo-Ternary Phase Diagram

Oleic acid is frequently utilized in topical preparations as an oil phase and a permeation enhancer [27]. In order to reduce skin irritation and system charge disruption, nonionic surfactants were chosen. PSM, the chosen surfactant in this study, has been previously used in transdermal formulations [28]. In addition, short-chain alcohols such as ethanol and propanol were included in the surfactant phase as cosurfactants to increase the microemulsion region in the phase diagrams. These alcohols have the ability to reduce the hydrophilicity of the polar solvent as well as solubilize high water contents and promote the formation of microemulsion [29,30]. Using a mixture of PSM and various cosurfactants (in ratios of 1:1 to 3:1, by weight) as the surfactant phase, oleic acid as the oil phase, and DI water as the aqueous phase, Figure 1 depicts the pseudo-ternary phase diagrams, illustrating the transparent microemulsion area.

Based on visual observation, the remaining portion of the phase diagram displayed turbidity and exhibited typical emulsion characteristics. The largest microemulsion area was produced by including 2-propanol as a cosurfactant, as shown in Figure 1. Furthermore, compared to formulations using ethanol as the cosurfactant, those using propanol had larger microemulsion areas. The results of the study indicate that the microemulsion area expanded as the chain length of the short-chain alcohols increased, progressing from ethanol to 2-propanol [31]. Additionally, no statistically significant difference was observed in the overall percentage of microemulsion area in the phase diagram when comparing formulations comprising 95% ethanol and DEGEE as the cosurfactant (*p* > 0.05).

### 3.3. Solubility of Miconazole Nitrate in Blank Microemulsions

The miconazole nitrate concentration used in the microemulsion formulations was decided based on its solubility in the different blank microemulsions. The values of equilibrium solubility are shown in Table 2, which demonstrates that the amounts of miconazole nitrate entrapped in blank microemulsions were significantly greater (*p* < 0.05) compared to DI water. The increased drug solubility in microemulsions can be explained by their ability to be solubilized within the interfacial film that forms between the water and oil phases. This interfacial film provides additional sites for the drug to dissolve, resulting in enhanced drug solubility. Based on the solubility study of miconazole nitrate in microemulsions, a concentration of 1% *w*/*w* was selected for the preparation of miconazole nitrate-loaded microemulsions.

### 3.4. Characterisation of Miconazole Nitrate-Loaded Microemulsions

All microemulsions, both unloaded and loaded with miconazole nitrate, were clear, yellowish liquids. They were optically isotropic because they lacked birefringence and appeared uniformly dark under a cross-polarized light microscope. A cross-polarized light microscope is a valuable tool for distinguishing between liquid crystals and microemulsions due to their visual similarity, particularly with lamellar and hexagonal liquid crystals. For lamellar and hexagonal liquid crystals, birefringence can be seen using a cross-polarized light microscope; however, microemulsions do not exhibit birefringence [32].

A dilution test was used to identify the formulation type. The results showed that a brilliant blue aqueous solution, but not oleic acid, could be used to dilute all o/w microemulsions. The formulation’s inclusion of miconazole nitrate did not alter the kind of microemulsion in any of the unloaded microemulsions. The formulations showed a conductivity of 7.5–101.5 μS/cm, indicating that the type of microemulsion formed was o/w (Table 3). Miconazole nitrate incorporation significantly increased the electrical conductivity of the unloaded microemulsions. The unloaded microemulsions had apparent pH values ranging from 4.6 to 5.5. All unloaded microemulsions’ pH values were slightly reduced by the addition of miconazole nitrate (*p* > 0.05) (Table 3).

Table 3 shows the apparent viscosity values for all microemulsions, both unloaded and loaded, at a shear rate of 1000 s^−1^ and a temperature of 30 ± 2 °C. Newtonian flow behavior was present in all microemulsion samples [33]. Miconazole nitrate incorporation had a minor impact on the microemulsion’s viscosity (*p* > 0.05), but it had no overall impact on the flow behavior. Unloaded microemulsions had typical droplet sizes varying between 49 and 372 nm. Each formulation’s droplet diameter was significantly impacted by the addition of miconazole nitrate (*p* < 0.05). In microemulsions containing miconazole nitrate, the average droplet size ranged from 54 to 404 nm (Table 3). Microemulsions typically have droplet sizes between 10 and 140 nm [34]. Larger sizes could, however, occur and be related to the dynamic characteristics of microemulsions. For instance, a microemulsion made of clove oil, polyoxyethylene sorbitan monolaurate, propylene glycol, water, and ketoprofen had mean droplet sizes of 396 nm and a highly variable size distribution [35]. It might be argued that because the creation of microemulsions requires negative Gibbs free energy, they can arise spontaneously, which results in high entropy and dynamic properties of the microemulsion system [36].

### 3.5. Thermodynamic Stability of Miconazole Nitrate-Loaded Microemulsions

Microemulsions are characterized by their thermodynamic stability, which sets them apart from emulsions that are kinetically stable and prone to phase separation over time [37]. None of the microemulsions, whether unloaded or loaded with miconazole nitrate, exhibited phase separation after the centrifugation test. Phase separation was observed only in the unloaded and miconazole nitrate-loaded microemulsions that contained PSM and DEGEE in a 3:1 ratio as the surfactant/cosurfactant after the third cycle of the freeze–thaw stress test. This indicates that these particular formulations were not thermodynamically stable, as indicated by the presence of phase separation.

### 3.6. Ex Vivo Skin Permeation Study of the Selected Miconazole Nitrate-Loaded Microemulsion

Most of the miconazole nitrate-loaded microemulsions exhibited good physicochemical properties. Among all formulations, the miconazole nitrate-loaded microemulsion containing PSM and ethanol in a 1:1 weight ratio as surfactant/cosurfactant was chosen for the ex vivo skin permeation study compared with the conventional cream due to its smaller size. The results of the drug permeation study of the microemulsion and conventional cream at 1% *w*/*w* miconazole nitrate are illustrated in Figure 2. At 24 h, the mean cumulative amounts of the drug permeated across pig skin were determined to be 87.6 ± 5.8 and 16.1 ± 2.9 μg/cm^2^ from the microemulsion and conventional cream, respectively. The permeation potential of the conventional cream was less than that of the microemulsion due to the semisolid cream base. The calculated permeation flux values were determined to be 5.4 ± 0.2 and 2.4 ± 0.1 μg/cm^2^/h for the microemulsion and conventional cream, respectively. The drug depositions from the microemulsion and conventional cream were 559.1 ± 25.1 and 284.9 ± 12.1 μg/cm^2^, respectively. Therefore, the selected miconazole nitrate-loaded microemulsion illustrated higher cumulative permeation, permeation flux, and drug deposition (retention) than the conventional cream at the end of 24 h. This would be attributable to the various components of microemulsion [38,39]. In addition, this nanometer-sized colloidal carrier increases drug penetration into the skin; the drug that has penetrated the skin concentrates there and stays localized for an extended period, facilitating targeted drug delivery to the skin [40].

### 3.7. In Vitro Antifungal Activity of the Selected Miconazole Nitrate-Loaded Microemulsion

Both acute and chronic cutaneous candidiasis are brought on by a type of Candida. Therefore, research into in vitro antifungal activities against these primary causative factors is necessary. The selected miconazole nitrate-loaded microemulsion and conventional cream revealed apparent zones of inhibition around the well against standard strain *C. albicans* ATCC 10231. The result revealed that there was a substantial increase in the in vitro inhibition of *C. albicans* by the microemulsion formulation (ZOI of 23.3 ± 1.5 mm) compared with the conventional cream (ZOI of 17.3 ± 0.6 mm) (*p* < 0.05) (Figure 3). The nanometer-sized particles and the high miconazole nitrate solubility in the presence of the surfactant and cosurfactant may be responsible for the increased antifungal activity of the microemulsion formulation. Nanometer-sized particles, owing to their larger surface area, exhibit enhanced permeation capabilities that allow them to easily penetrate the cell walls of fungal strains [41].

### 3.8. Stability Study of the Miconazole Nitrate-Loaded Microemulsion

The optimal miconazole nitrate-loaded microemulsion formulation demonstrated good physicochemical stability at room temperature for 3 months. There was no physical appearance change between t = 0 and t = 3 months and a remaining percentage of miconazole nitrate greater than 95.0% in the formulation.

### 3.9. Analysis of Miconazole Nitrate by HPLC

The linear equation for the calibration curve of miconazole nitrate was y = 1167.3x − 189.23, demonstrating a strong linear correlation coefficient of 1. The %RSDs of intra-day and inter-day precision were less than 2.0%. The mean recovery for accuracy determination was 100.4 ± 0.3%. The LOQ and LOD values for miconazole nitrate were 0.51 and 0.17 μg/mL, respectively.

### 3.10. NIR Spectroscopic Measurement Combined with PLSR Model

The development of a non-destructive NIR spectroscopic analysis for the quantitative determination of miconazole nitrate-loaded microemulsions is an innovative and significant step towards the efficient and reliable analysis of these formulations without requiring sample preparation. The microemulsion contains an infinitesimal amount of miconazole nitrate content, which posed a challenge to the development of the NIR analysis.

Several PLSR models were performed with both raw and pretreated NIR data, with respect to the actual values acquired from the HPLC method. Various parameters, including the R^2^ model, R^2^ Pearson, RMSEP, RMSEC, and bias, were utilized to choose the optimum model. The R^2^ model ranges between 0 and 1, where a value of 1 indicates a perfect fit of the model to the data, and a value of 0 suggests no relationship between the model and the data and is used to assess the model’s goodness-of-fit. A high value for the R^2^ Pearson denotes a good fit between the predicted and actual values. RMSEP measures the model’s accuracy, with a small value implying good predictive ability. A good PLSR model should have a low RMSEC, indicating good accuracy in predicting the calibration set. Finally, a low bias indicates an unbiased and accurate model between the predicted and actual values.

The results indicate that the most appropriate PLSR model was obtained using the OSC pretreatment method, as it yielded a high R^2^ model of 0.9919 and a high R^2^ Pearson of 0.9958. Additionally, it had the lowest RMSEC of 0.0488 and RMSEP of 0.0390 among all the pretreatment methods, suggesting good predictive power and generalizability to new samples. The relatively low bias value of 0.0061 further indicates that there is no significant systematic bias in the predictions (Table 4 and Figure 4). Although a good PLSR model could be derived from the raw data without any pretreatment methods (model 1 in Table 4), OSC data pretreatment improved model accuracy and prediction efficiency. Furthermore, the principal component analysis of all the spectral data matrix with OSC pretreatment showed better sample grouping by miconazole nitrate concentrations than the original spectral data without pretreatment (Figure 5). The 1D, 2D, SNV, and area normalization data pretreatment could also provide acceptable PLSR models, but most of these were not dramatically improved in the R^2^ model, R^2^ Pearson, RMSEC, RMSEP, and bias compared with the original data and OSC pretreated models. These findings suggest that OSC pretreatment is the most appropriate method for deriving the PLSR model with high accuracy and prediction efficiency, as well as enhancing the separation and discrimination of the sample groups.

## 4. Conclusions

Miconazole nitrate has been a promising antifungal agent for treating superficial and deep skin infections, primarily caused by Candida strains. However, its limited water solubility has posed a significant challenge for developing effective formulations that can penetrate the skin topically. To enhance topical delivery, miconazole-loaded microemulsions were prepared via spontaneous emulsification. In vitro and ex vivo experiments were conducted to assess the potential of a 1% *w*/*w* miconazole nitrate-loaded microemulsion containing oleic acid as an oil phase, PSM and ethanol (1:1) as a surfactant/cosurfactant system, and DI water as an aqueous phase. The results indicate that this microemulsion could effectively deliver miconazole nitrate topically and enhance its bioavailability. Moreover, the study explored the feasibility of using NIR spectroscopy combined with the PLSR model to quantify miconazole nitrate in microemulsion systems. The findings demonstrated that there were no significant statistical differences between the results obtained from the proposed and HPLC methods. The method could potentially be applied to monitor the quality control of miconazole nitrate in various conventional and novel formulations. Furthermore, it offers a rapid, non-destructive analysis without requiring sample preparation.

## Figures and Tables

**Figure 1 pharmaceutics-15-01637-f001:**
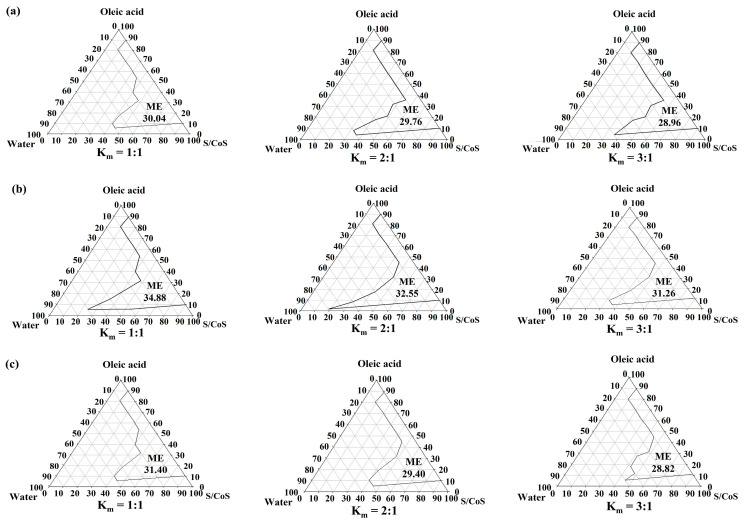
Pseudo-ternary phase diagrams of microemulsions containing oleic acid as the oil phase. (**a**) represents the phase diagrams of microemulsions with polyoxyethylene sorbitan monooleate (PSM) as a surfactant (S) and ethanol as a cosurfactant (CoS) (PSM/ethanol, S/CoS); (**b**) represents the phase diagrams of microemulsions with PSM as a surfactant and 2-propanol as a cosurfactant (PSM/propanol, S/CoS); (**c**) represents the phase diagrams of microemulsions with PSM as a surfactant and 2-(2-ethoxyethoxy) ethanol (DEGEE) as a cosurfactant (PSM/DEGEE, S/CoS). The weight ratios of surfactant to cosurfactant are represented by K_m_. The percentage of the microemulsion (ME) regions in the overall phase diagrams is indicated by the numbers within the respective areas.

**Figure 2 pharmaceutics-15-01637-f002:**
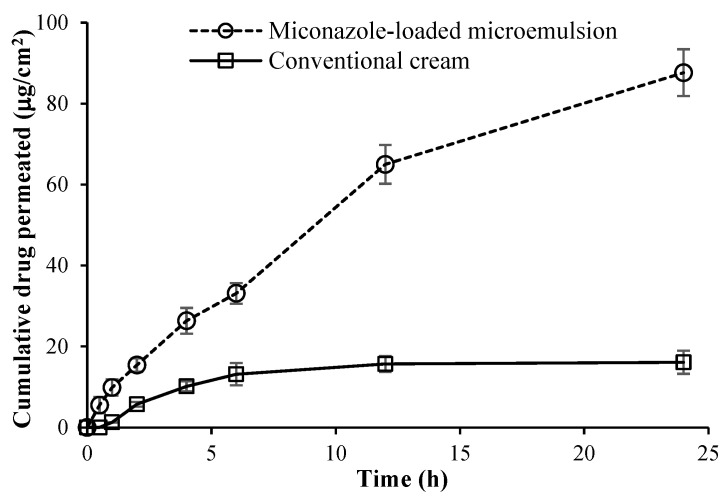
Permeation study across pig skin placed for the miconazole nitrate-loaded microemulsion containing polyoxyethylene sorbitan monooleate and ethanol at a weight ratio of 1:1 as surfactant/cosurfactant and the conventional cream at 1% *w*/*w* miconazole nitrate using vertical Franz diffusion cells (cumulative drug permeation). The experiment was done at least three times, using skin samples from a minimum of three newborn pigs.

**Figure 3 pharmaceutics-15-01637-f003:**
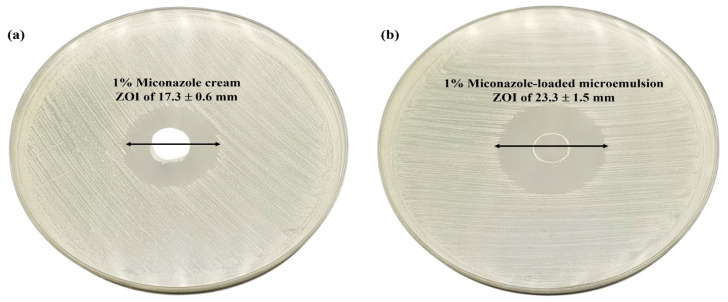
In vitro anticandidal activity of (**a**) 1% *w/w* miconazole cream and (**b**) 1% *w*/*w* miconazole-loaded microemulsion using the cup-plate method. The experiment was done in triplicate.

**Figure 4 pharmaceutics-15-01637-f004:**
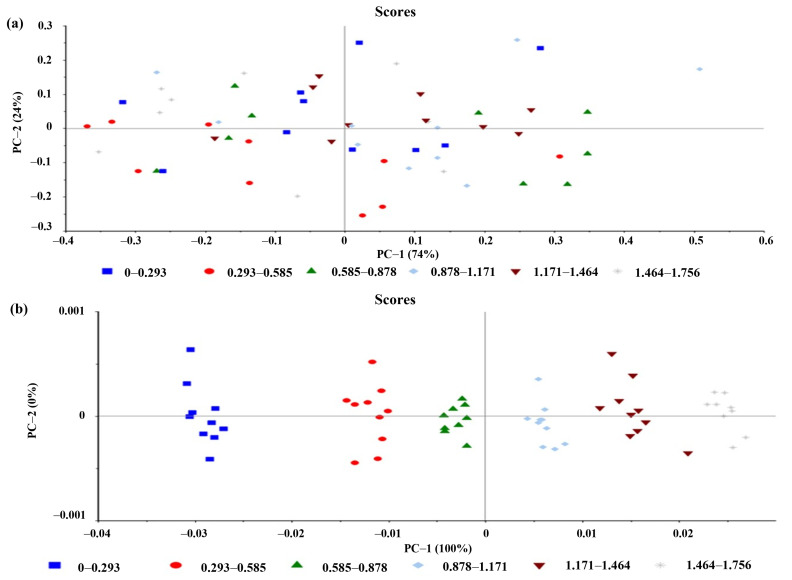
(**a**) Score plot of original near-infrared spectral data without any pretreatment showed ungrouping of samples along with PC1 and PC2 and (**b**) after orthogonal signal correction pretreatment, the plot showed better sample grouping by concentrations compared with data without pretreatment.

**Figure 5 pharmaceutics-15-01637-f005:**
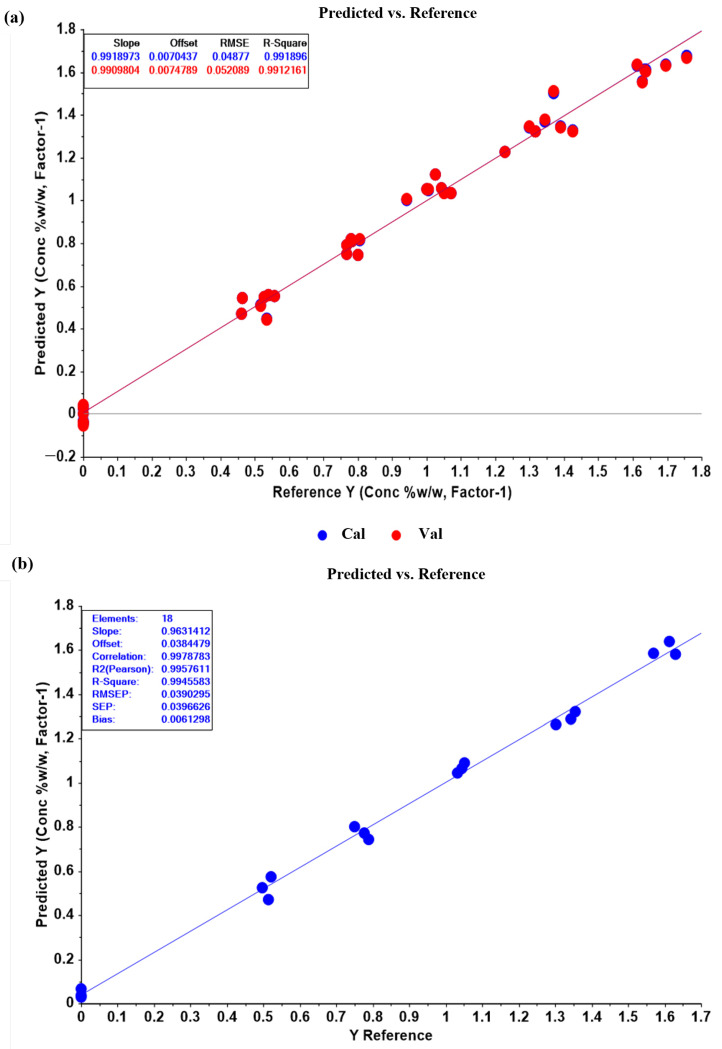
(**a**) Plot of the partial least-squares regression (PLSR) calibration model of orthogonal signal correction (OSC) pretreated data. The model showed consistency in calibration and cross-validation plots (blue line and red line). Slope, offset (intercept), root mean square error (RMSE) and R^2^ of the two lines were close to each other, indicating the accuracy of the model. (**b**) Plot of prediction results obtained from the PLSR model of OSC pretreated data. The prediction plot of miconazole concentrations in the validation set showed a high correlation between the concentrations obtained from the PLSR model (y-axis) and HPLC values (x-axis), with an R^2^ Pearson of 0.9958 and low error in terms of RMSEP and bias.

**Table 1 pharmaceutics-15-01637-t001:** Miconazole nitrate’s equilibrium solubility in the tested solvents at a temperature of 30 ± 2 °C.

Phase	Excipient	Equilibrium Solubility ^a^ (% *w*/*w*)
Oil	Oleic acid	0.012 ± 0.003
	Caprylic/capric triglyceride	0.007 ± 0.003
Surfactant	Polyoxyethylene sorbitan monooleate	0.654 ± 0.066
	Sorbitan monooleate	0.433 ± 0.095
Cosurfactant	Ethanol	1.164 ± 0.026
	2-(2-Ethoxyethoxy) ethanol	0.965 ± 0.131
	2-Propanol	0.225 ± 0.052

^a^ Values are shown as mean ± SD (*n* = 3).

**Table 2 pharmaceutics-15-01637-t002:** Equilibrium solubility of miconazole nitrate in the blank microemulsions at 30 ± 2 °C.

Sample	Equilibrium Solubility ^a^ (%*w*/*w*)
Blank microemulsions	
PSM:Ethanol (1:1)	1.560 ± 0.009
PSM:Ethanol (2:1)	1.450 ± 0.050
PSM:Ethanol (3:1)	1.062 ± 0.101
PSM:2-Propanol (1:1)	1.237 ± 0.025
PSM:2-Propanol (2:1)	1.290 ± 0.032
PSM:2-Propanol (3:1)	1.232 ± 0.095
PSM:2-(2-Ethoxyethoxy) ethanol (1:1)	1.384 ± 0.033
PSM: 2-(2-Ethoxyethoxy) ethanol (2:1)	1.069 ± 0.241
PSM: 2-(2-Ethoxyethoxy) ethanol (3:1)	1.068 ± 0.150
Deionized water	0.026 ± 0.001

PSM = polyoxyethylene sorbitan monooleate; ^a^ Values are shown as mean ± SD (*n* = 3).

**Table 3 pharmaceutics-15-01637-t003:** Physicochemical characterizations of all unloaded and miconazole nitrate-loaded microemulsions.

Formulations	Electrical Conductivity ^a^ (μS/cm)	pH ^a^	Apparent Viscosity at 1000 s^−1 a^ (mPa.s)	Mean Particle Size ^a^ (nm)	Polydispersity Index ^a^
Unloaded o/w microemulsions					
1. PSM:Ethanol (1:1)	16.65 (0.02)	4.7 (0.0)	38.49 (4.36)	49 (1)	0.26 (0.08)
2. PSM:Ethanol (2:1)	21.80 (0.01)	5.0 (0.0)	98.73 (18.15)	137 (3)	0.31 (0.01)
3. PSM:Ethanol (3:1)	13.13 (0.01)	5.3 (0.0)	132.34 (5.25)	251 (8)	0.49 (0.04)
4. PSM:2-Propanol (1:1)	8.00 (0.02)	5.3 (0.0)	30.96 (1.88)	70 (1)	0.42 (0.12)
5. PSM:2-Propanol (2:1)	7.54 (0.03)	5.5 (0.0)	73.67 (0.45)	181 (1)	0.48 (0.10)
6. PSM:2-Propanol (3:1)	9.26 (0.01)	5.1 (0.0)	107.28 (2.82)	305 (8)	0.60 (0.14)
7. PSM:2-(2-Ethoxyethoxy) ethanol (1:1)	11.85 (0.01)	4.6 (0.0)	61.16 (0.33)	200 (4)	0.46 (0.06)
8. PSM:2-(2-Ethoxyethoxy) ethanol (2:1)	11.52 (0.03)	4.7 (0.0)	117.34 (8.17)	287 (8)	0.51 (0.10)
9. PSM:2-(2-Ethoxyethoxy) ethanol (3:1)	10.52 (0.03)	4.7 (0.0)	169.76 (8.32)	372 (9)	0.59 (0.16)
1% *w/w* Miconazole nitrate-loaded o/w microemulsions					
1. PSM:Ethanol (1:1)	101.50 (0.01)	4.6 (0.0)	40.37 (8.17)	54 (2)	0.22 (0.10)
2. PSM:Ethanol (2:1)	72.50 (0.02)	4.7 (0.0)	84.14 (4.66)	156 (7)	0.31 (0.11)
3. PSM:Ethanol (3:1)	55.60 (0.02)	4.7 (0.0)	133.52 (4.62)	275 (12)	0.35 (0.20)
4. PSM:2-Propanol (1:1)	66.80 (0.01)	4.6 (0.0)	26.27 (2.91)	73 (1)	0.48 (0.06)
5. PSM:2-Propanol (2:1)	54.70 (0.03)	5.0 (0.0)	67.11 (7.68)	188 (4)	0.50 (0.07)
6. PSM:2-Propanol (3:1)	47.00 (0.00)	5.0 (0.0)	102.74 (10.03)	328 (13)	0.61 (0.13)
7. PSM:2-(2-Ethoxyethoxy) ethanol (1:1)	55.20 (0.00)	4.5 (0.0)	59.78 (4.67)	208 (1)	0.46 (0.03)
8. PSM:2-(2-Ethoxyethoxy) ethanol (2:1)	43.00 (0.00)	4.6 (0.0)	111.28 (8.02)	318 (18)	0.40 (0.08)
9. PSM:2-(2-Ethoxyethoxy) ethanol (3:1)	34.50 (0.00)	4.4 (0.0)	185.36 (7.38)	404 (14)	0.47 (0.04)

^a^ Mean of triplicate experiments; numbers in parentheses represent SD values; PSM = polyoxyethylene sorbitan monooleate.

**Table 4 pharmaceutics-15-01637-t004:** Constructed PLSR models with model parameters.

Data Pretreatment	Factors	R^2^ Model	RMSEC	R^2^ Pearson	RMSEP	Bias
Original	9	0.9845	0.0674	0.9861	0.0696	0.0106
First derivative	6	0.9844	0.0676	0.9790	0.0775	0.0042
Second derivative	6	0.9882	0.0589	0.9607	0.1122	−0.0237
Standard normal variate	5	0.9672	0.0980	0.9633	0.1074	−0.0149
Area normalization	6	0.9661	0.0997	0.9599	0.1118	−0.0314
Orthogonal signal correction	1	0.9919	0.0488	0.9958	0.0390	0.0061

RMSEC = root mean square error of calibration; RMSEP = root mean square error of prediction.

## Data Availability

The data presented in this study can be obtained by contacting the corresponding author and requesting access to the data.

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
