# Peer review of "Development of Miconazole-Loaded Microemulsions for Enhanced Topical Delivery and Non-Destructive Analysis by Near-Infrared Spectroscopy"

_pharmaceutics, 2023, doi:10.3390/pharmaceutics15061637_

Round 1
Reviewer 1 Report
The authors should clarify the following:
- In the section 2. Materials and Methods, subsection 2.4, lines 172-178, the authors should explain the preparation of the tested microemulsions by loading the blank microemulsions with miconazole nitrate instead of dissolving the drug in the oil-surfactant-cosurfactant mixture prior adding the aqueous phase.
- In the section 2. Materials and Methods, subsection 2.7, lines 230-231, the authors should explain the reason for using methanol as cosolvent in the ex vivo skin permeation study, considering its toxicity at skin level and its ability to alter skin permeability.
In the section 3 Results and Disscusion, subsection 3.2, lines 376-378, the authors should reformulate the phrase, as cosurfactants do not “aid in the formation of phase diagrams”. Suggestion: cosurfactants can increase the microemulsion region in the phase diagrams.
Author Response
Thank you very much for the review of our manuscript. We sincerely appreciate all the valuable comments and suggestions, which helped us to improve the quality of the article. Our responses to the reviewer's comment are described in a point-to-point manner, as shown in the attached file.

Reviewer 2 Report
In this paper, miconazole-loaded microemulsions were developed and assessed for topical skin delivery. The authors systematically optimized the preparation of the miconazole-loaded microemulsions. Besides, the study explored the feasibility of using NIR spectroscopy combined with the PLSR model to quantify miconazole nitrate in microemulsion systems. Overall, this paper is well designed and the results are interesting, which could be considered for publication.
Author Response
The authors would like to thank Reviewer 2 for taking the necessary time and effort to review our manuscript.
Reviewer 3 Report
The paper is very interesting however, there are several errors which have to be corrected before it may be accepted for publication:
1. The figures should be self explanatory - all their elements have to be described in the caption. Please change to make it easier for the reader to understand (Fig. 3, 4).
2. In the results section, we have the descriptions of the experiments, which should be moved to materials and methods section (par. 3.8, 3.9).
3. I belive that adding the detailed description or scheme of pig skin expeiments should make it easier to reproduce by the readers.
4. Please add the information about the ethical committee localization.
5. Fig. 4A is enlarged in x-axis and the whole plot looks bad - please move to the original sizing of the graph or enlarge it proportionally in x- and y-axes.
Otherwise the paper is valuable.
Author Response

(The authors gave the same response as above.)

Reviewer 4 Report
The aim of the work was to develop a new formulation (microemulsion), with increased availability of miconazole nitrate. Based on a wide range of studies (physical appearance, conductivity and pH measurements, particle size, viscosity and rheological behavior, stability, skin permeation, antifungal activity), the authors showed that the developed new formulation is characterized by higher cumulative permeation, permeation flux and drug deposition compared to 1% conventional cream with miconazole nitrate. They also found a significant increase in in vivo inhibition of C. albicans.
An additional element of the work was also the development of a new, chemometric method of quantitative analysis of miconazole nitrate in microemulsion based on NIR spectra. Calculations made with PLSR showed that data on the content of miconazole nitrate in microemulsion are closely correlated with results obtained with HPLC. The newly developed chemometric method can therefore be used for quantitative analysis of microemulsion, especially due to the fact that it is a non-destructive method.
Before publishing the work, the authors should clarify some issues.
lines 266-268 – the sentence “The SDA medium was prepared by weighing the required amount and dissolving it in water, followed by sterilization in an autoclave at 121°C for 15 min.” is incomplete. It does not say what substance was weighed.
Section 2.11.2. – there is no information on the spectral range in which NIR spectra were recorded. What data from NIR spectra were used to construct the matrix for the PLSR calculations?
line 549 – PCA was also used in the chemometric analysis. Therefore, section 2.11.3. should contain information on how was the matrix for the PCA calculations constructed? Additionally, please explain whether the PCA was performed on a covariance or a correlation matrix?
Author Response

(The authors gave the same response as above.)

Reviewer 5 Report
In the authors' current work, microemulsions were used as transdermal delivery vehicles for the antifungal drug miconazole nitrate with a view to improving the solubility and transdermal permeability of the drug, and the results met expectations. A method has also been developed for the determination of drug in products by near infrared spectroscopy, which has the advantage of being performed without destroying the sample. The work can be accepted subject to the following issues being addressed.
1. There are some problems with the writing format, such as negative signs, and there is a mix of - and ‒. Please check the full text and correct any writing errors that exist.
2. Experimental animals should be provided with the ethical approval number.
3. How was the hair removed from the isolated skin?
4. Please convert the centrifugal speed to centrifugal force for separation by centrifugation, otherwise the centrifugal radius should be provided.
5. Please add data processing methods and statistical analysis methods.
6. Please indicate the number of repetitions of each experimental item. The number of samples (n=?) should be provided in the title of the corresponding figures and tables. .
7. In vitro antifungal activity test results would be of more interest to the reader if corresponding photographs were shown.
8. Some reports on microemulsion as a transdermal drug delivery vehicle (Drug Deliv, 2021,28(1):2062-2070, doi: 10.1080/10717544.2021.1983073; J Nanobiotechnology, 2018,16(1):91, doi: 10.1186/s12951-018-0418-2) would help to enrich the knowledge of microemulsion and are recommended to be cited in future revisions.
Minor editing of English language required.
Author Response

(The authors gave the same response as above.)
